# Comprehensive Gerontological Assessment: An Update on the Concept and Its Evaluation Tools in Latin America and the Caribbean—A Literature Review

**DOI:** 10.3390/ijerph21121697

**Published:** 2024-12-19

**Authors:** Rafael Pizarro-Mena, Elena S. Rotarou, Diego Chavarro-Carvajal, Patrick Alexander Wachholz, María Fernanda López, Cristina Perdomo Delgado, Solange Parra-Soto, Isabel Barrientos-Calvo, Felipe Retamal-Walter, Gloria Riveros-Basoalto

**Affiliations:** 1Facultad de Odontología y Ciencias de la Rehabilitación, Universidad San Sebastián, Sede Los Leones, Santiago 7500000, Chile; cristina.perdomo@uss.cl; 2Red Interuniversitaria de Envejecimiento Saludable de Latinoamérica y Caribe (RIES-LAC); elena.rotarou@uss.cl (E.S.R.); diegoandreschavarro@gmail.com (D.C.-C.); patrick.wachholz@unesp.br (P.A.W.); licmflopez@gmail.com (M.F.L.); sparra@ubiobio.cl (S.P.-S.); isacrisba80@gmail.com (I.B.-C.); 3Núcleo Milenio Estudios en Discapacidad y Ciudadanía—DISCA (NCS2022_039), Santiago 7500000, Chile; 4Facultad de Medicina y Ciencia, Universidad San Sebastián, Sede Los Leones, Santiago 7500000, Chile; 5Instituto de envejecimiento de la Facultad de Medicina, Pontificia Universidad Javeriana, Bogotá 110231, Colombia; 6Unidad de Geriatría, Hospital Universitario San Ignacio, Bogotá 110231, Colombia; 7Faculdade de Medicina de Botucatu, Universidade Estadual Paulista—Unesp, São Paulo CEP 18618-687, Brazil; 8Facultad de Psicología y Ciencias Sociales, Universidad Flores, Buenos Aires C1406EEE, Argentina; 9Departamento de Nutrición y Salud Pública, Facultad Ciencias de la Salud y de los Alimentos, Universidad del Bio-Bio, Chillán 3780000, Chile; 10Unidad de Investigación Hospital Nacional de Geriatría y Gerontología, Universidad de Costa Rica, San José 11501-2060, Costa Rica; 11School of Health and Rehabilitation Sciences, The University of Queensland, Brisbane, QLD 4072, Australia; f.retamalwalter@uq.edu.au; 12Bibliotecóloga Referencista, Vicerrectoría Académica, Universidad San Sebastián, Santiago 7500000, Chile; gloria.riveros@uss.cl

**Keywords:** older adults, evaluation, Comprehensive Gerontological Assessment, Latin America, geriatric syndrome

## Abstract

In recent decades, Latin America and the Caribbean region have experienced accelerated aging. However, despite the significant progress of gerontology in the region, the number of geriatricians and professionals trained in the field is low; a notable gap in the training related to the assessment of older adults can also be observed. Therefore, in this literature review, we update the concept of Comprehensive Gerontological Assessment (CGA) for its application in the region. We compile the characteristics, validity, and reliability of CGA tools, and their usage in government documents across countries in the region. We also analyze the adaptations made to CGA during the COVID-19 pandemic, and discuss challenges related to CGA administration, academic training, research, innovation, and management. This study is particularly relevant as it proposes lines of action for decision-makers, academics, researchers, university students, and the general community, which will allow for more tailored interventions aimed at meeting the needs of older adults, their families, and caregivers. Such actions will positively impact functionality, autonomy, and quality of life, while promoting healthy, active, and successful aging in the region.

## 1. Introduction

Globally, the population aged 60 years and older is expected to increase from 900 million in 2015 to over 1.4 billion by 2030, representing a 64% rise in just 15 years, making it the fastest-growing age group [1]. In Latin America and the Caribbean, although the aging process is still incipient or moderate in half of the countries (many of which are developing nations), the region is entering a phase of accelerated aging, with the most significant changes expected by 2030 [1]. This rise in the proportion of older adults (OA) is occurring alongside an increase in chronic diseases, geriatric syndromes, disability, and dependency [2]. These demographic and epidemiological transitions have created a new healthcare reality, leading to increased needs and intervention opportunities for the older population in the region [2].

In this context, the science dedicated to studying aging and the elderly population is gerontology [3], which addresses all aspects of OA, their families, and their environment, drawing from multiple and diverse disciplines and professions [3]. Geriatrics, on the other hand, is the branch of this science focused on addressing the potentialities, needs, problems, and geriatric syndromes of OA at different levels of healthcare [4]. Both gerontology and geriatrics have experienced slow and gradual growth in the region, lagging behind the rapid demographic and epidemiological transitions [2]. This means that, to date, few undergraduate academic programs (for health and psychosocial professionals) include subjects or training content related to these fields, and they are almost non-existent in areas such as law, engineering, design, and architecture [2].

At the same time, the number of geriatricians, as a medical specialty, and other professionals with specialized training in the field is still very low in the countries of the region [2,3], with a greater presence in more aged countries such as Mexico, Costa Rica, Brazil, Colombia, Uruguay, Argentina, and Chile. Additionally, many doctors and professionals who already work with OA did not receive undergraduate training, resulting in a persistent educational gap [2]. Consequently, these professionals lack the knowledge, skills, and competencies related to key aspects of the OA care, such as geriatric syndromes and Comprehensive Gerontological Assessment (hereinafter, CGA) [5,6]. Furthermore, there are few opportunities for continuing education, specialization, or postgraduate training in these areas across the region [2].

CGA application aims to go beyond the traditional medical model by considering OA in all their complexity [7]. It has been shown to effectively reduce mortality, and complications, while enhancing functionality, cognitive and emotional states, and the accuracy of diagnoses. Additionally, CGA optimizes the use of programs and services for OA, reduces healthcare costs, and increases patient satisfaction, improving the likelihood that OA remain in their homes and reducing hospital admissions and readmissions [6,7,8,9,10], as well as polypharmacy at discharge [11]. Furthermore, CGA guides multicomponent interventions for OA [8].

Globally, CGA has expanded beyond hospital settings to primary healthcare (PHC), communities, and long-term care facilities for OA [12]. Various CGA assessment tools have been documented, for example, in Spain [8], where the most frequently used tools have been outlined [5]; in other cases, only specific assessment tools have been analyzed [9]. However, to date, there has been no comprehensive update on the validated and most commonly used CGA tools in the Latin American and Caribbean region.

Various CGA definitions underline the necessity of the assessment tools being validated and reliable [8,9]. Validation and reliability are key for the assessment process, since the needs and problems evaluated in OA are complex, requiring measurements that assign numerical value to subjective traits. However, a major limitation of CGA is the lack of standardization in both assessment and intervention [7,12]. Therefore, to obtain valid and reliable results, improve the efficiency of assessments, and ensure comparability, it is essential to standardize these tools and determine cut-off points based on local realities [13].

Validity (accuracy) refers to the degree to which a measurement reflects the true nature of a phenomenon, assessing whether the instrument or method accurately measures or classifies what it is intended to [14]. There are three types of validity: construct, content, and criterion. Criterion validity includes concurrent and predictive validity [14]. On the other hand, reliability (precision) relates to the consistency of measurement scores, indicating how free they are from measurement error, that is, when measurements are repeated under constant conditions, they should yield similar results [14]. Reliability is linked to the instrument’s stability, regardless of the person that applies it (observer) or timing of the application [14,15]. Various methods and statistics are used to analyze reliability [14,15].

Therefore, CGA tools’ validity and reliability should converge to obtain greater confidence in the results [15]. In the CGA context, and after decades of aging in the region, it is relevant to identify the tools that have undergone transcultural adaptation (since they typically come from another language, usually English), and determine their validity and/or reliability. Furthermore, it is useful to identify the tools that have been designed and/or validated in countries in the region, as well as public policies that have included CGA, including the tools they use and/or recommendations made in their government documents.

In addition to the above, the COVID-19 pandemic generated a series of negative effects on the health of OA, even being described as a geriatric emergency [16], exacerbating some existing problems and bringing others to light. The pandemic, at the same time, emerged as an opportunity to implement telehealth with OA [17,18]. Telehealth refers to a modality that uses information and communication technologies (ICTs) to provide health services, medical care, and information, regardless of distance [19]; it also includes telemedicine and telerehabilitation as specific modalities [18,19,20]. This led to CGA evaluation and intervention through telehealth, providing continuity of care for OA during the pandemic [17]. Furthermore, prior to the pandemic, there were already experiences of the usage of validated CGA assessment tools for telephone use [21], which could offer an opportunity for innovation in interventions with OA through telehealth, during the pandemic and beyond. The CGA adaptations that were implemented during the pandemic and involved the use of telehealth could provide valuable lessons for employing this modality of care in regular and non-emergency situations as well.

On the one hand, the local reality in the region reveals the growing and multiple intervention needs of OA, the lack of doctors and specialized professionals, the lack of training of professionals already working with OA on CGA issues, and the lack of undergraduate and postgraduate training in gerontology and geriatrics. On the other hand, the increasing advancement of research regarding CGA and its assessment tools (pre- and post-pandemic), and the need for CGA standardization in the region create an opportunity for updating in this field. Therefore, this literature review aims to: (a) conceptualize Comprehensive Gerontological Assessment for its use in the Latin America and Caribbean region; (b) analyze the characteristics of the validated and most commonly used CGA tools in the region; (c) analyze the adaptations made to CGA during the COVID-19 pandemic in the region; (d) discuss challenges and opportunities regarding the administration, academic training, research, innovation, and management of CGA in the region, which will lead to an improvement in CGA management by doctors and professionals, and enhance the quality of life for OA.

## 2. Materials and Methods

### Search Strategy

A literature review was chosen as it allows for a broader and more diverse identification of available literature, as well as for more flexible and contextual exploration [22]. To achieve the aims of this literature review [22], a literature search was conducted to answer the question: *How were the Comprehensive Gerontological Assessment (CGA) and its assessment tools implemented in Latin America and the Caribbean before, during, and after the COVID-19 pandemic?* The keywords and search terms (Mesh/DeCS), in both English and Spanish, were: older adult, older person, older people, geriatric assessment, comprehensive gerontological assessment, comprehensive geronto-geriatric assessment, comprehensive geriatric assessment, tool, instrument, validation, reliability.

The databases consulted were PubMed, Scopus, and Scielo, and included articles published in Spanish, Portuguese, and English. Google Scholar and government documents from the countries of Latin America and the Caribbean were also consulted. The search was conducted from the year 2000 onwards.

Scientific articles were included if they contributed to answering the objectives of the literature review in conceptual and contextual terms or if they determined the validity, reliability, and/or provided reference values for any of the assessment tools in the countries of Latin America and the Caribbean. This could involve studies focusing exclusively on populations aged 60 years and older or as part of the general population.

Additionally, government documents and technical guidelines from countries in the region were included, whether they focused specifically on older populations or the general population. These documents needed to include any of the tools identified as valid and/or reliable in this review or propose CGA batteries for older adults at different levels of care, programs, and/or social-health services for older adults in countries in Latin America and the Caribbean region.

Abstracts from conferences, scientific congress reports, dissertations, research reports, theses, letters to the editor, editorials, and commentaries were excluded.

The following steps were followed in the literature review process [23], depending on the type of document. In the case of scientific articles, the steps were: strategy-based search, article identification, duplicate elimination, title and abstract screening, full-text reading, selection, compilation into tables, analysis, interpretation of results and discussion, and referencing. Using the list of assessment tools for which validity, reliability, and/or reference values were determined, the review process proceeded to government documents from countries in the region. In the case of government documents, the steps were: search, identification, full-text reading focused on identifying the selected tools and CGA battery proposals mentioned in the document, selection, compilation into tables, analysis, interpretation of results and discussion, and referencing. A reference librarian, part of the research team, collaborated throughout the various stages of the literature review process.

A total of 800 scientific articles and 40 government documents were initially identified for the literature review. There were 254 duplicates, which were eliminated. After applying inclusion and exclusion criteria, 129 documents were finally selected for the literature review (110 scientific articles and 19 government documents). Figure 1 provides a diagram of the article selection process for the literature review.

The information extracted from the studies and documents included in this review was synthesized by the authors to specifically address each objective of the present study.

## 3. Results

### 3.1. Conceptualization of Comprehensive Gerontological Assessment: A Current and Expanded Perspective

Traditionally, from the field of geriatrics and through the expertise of geriatricians, the concept of Comprehensive Geriatric Assessment was initially proposed and used. This type of assessment—with its classic four domains: biomedical or clinical, functional, mental, and socio-familial [5,6]—has a biomedical, reductionist, and geriatrized view of the social and healthcare services for older adults. This concept has got closer to gerontology (with a biopsychosocial, broad, and holistic vision) and has transitioned to the concept of Comprehensive Geronto-Geriatric Assessment [24,25,26], which has already been used in some public policy programs [24,25,26]. However, considering the geographic, demographic, and ethnic diversity (from north to south, from coast to mountain ranges, from plains to valleys, from continent to islands) that characterizes Latin America and the Caribbean, the significant progress in gerontology [3], the diversity of university professions that address OA in the region (some of which are unique and present only in certain countries, such as gerontologists, psychometricians, and music therapists), and the variety of contexts in which interventions with OA take place (ranging from the hospital to the community, urban and rural), this concept should definitively transition to the concept of Comprehensive Gerontological Assessment, a proposal which we make in this literature review. CGA should also be understood as one of the three main pillars or cornerstones of gerontology and geriatrics, along with interdisciplinary work and level-based coordination and management [27].

This concept is not new and has been used before, for example, at the XXV International Congress of the Galician Society of Gerontology and Geriatrics in 2013 in Spain [28]. Furthermore, it has been incorporated into postgraduate programs aimed at physicians, health professionals, and psychosocial workers, such as the Master’s in Gerontological and Geriatric Kinesiology at Universidad San Sebastián in Chile, and the Doctorate in Gerontological Research at Universidad Maimónides in Argentina. Additionally, it has been used in recent scientific articles in both Spanish [3,29,30] and English [31,32], and in resolutions by the Ministry of Health of Argentina [33].

CGA is a technology within gerontology and geriatrics [4], which we conceptualize as ‘the interdisciplinary diagnostic process aimed at identifying and/or quantifying the needs, problems, potentialities, and opportunities that older adults and their environment present in biomedical, physical-functional, cognitive, affective, socio-familial, environmental, and quality-of-life aspects, with the goal of developing promotional, preventive, therapeutic, rehabilitative, and palliative care and a follow-up care plan that is integrated and coordinated to meet these needs in order to achieve maximum autonomy, functionality, and quality of life for the older adult’ [5,6,31]. This diagnostic process emphasizes the early detection of pre-frail and frail OA with multimorbidity for the prevention of geriatric syndromes, disability, and dependency, and serves as an admission protocol for any institution [4,27,34,35].

CGA includes a set of assessment tools (objective, specific, valid, reliable, standardized, reproducible, simple, and low-cost) for these domains and spheres of OA’s lives [8,9], and employs scales, questionnaires/surveys, indexes, clinical and performance tests, and schematics/drawings/pictograms, which are used in various contexts, levels of care, programs, and services involving OA and their environment [31].

The assessment made with these tools needs to be accompanied, on the one hand, by an interview with the OA, their family member, and/or caregiver [36], and, on the other hand, by clinical evaluations and the expertise of each profession, identifying quantitative polypharmacy and potentially inappropriate medication, as well as laboratory and imaging tests aimed at delving into the different CGA domains and spheres [24,27,37]. Conducting CGA and addressing these domains requires specialized training and time [5,8], which may vary from several sessions [27] to a process lasting up to two weeks, as described in some programs [24]. All the information gathered should be recorded in a geronto-geriatric registration form, as seen in primary healthcare in Mexico [38], and day centers [26] and residences for OA in Chile [24].

Ideally, this diagnostic process should continue with the presentation of the assessments conducted by each professional in the interdisciplinary team meeting [8,24], with the aim of defining general, specific, and discipline-specific intervention objectives, timelines, intervention and prognoses [24,27]. At the same time, it is important to prioritize which factors, diseases, or geriatric syndromes should be addressed first in the OA; these are usually the conditions that pose the greatest health risk, are triggering factors, or have the most significant impact on OA’s quality of life and survival [27]. Consequently, the CGA as a process, includes the evaluation and the definition of the intervention plan, which is a strength [5].

It is important to include OA in the CGA process, by actively involving them in their assessment and in the definition of objectives, prognoses, and the intervention plan [24,27], considering their expectations, needs, fears [24], preferences, and values [5], as well as their context, and following the principles of the person-centered care model [24,39], which permeates gerontology throughout this process.

This is how CGA allows for the systematization of information, the establishment of a common language [5,8], the unification of criteria and diagnoses, and the objective evaluation of the problems or conditions that OA present, enabling proper follow-up. It also allows for the evaluation of the intervention and reduces assessment time, as its use provides a general and comprehensive view of OA’s health status [5].

Consequently, it is necessary to move towards a vision of CGA from the perspective of gerontology, considering the progression of aging and the diverse characteristics of OA in the countries of the region. It is important to go beyond its current conceptualizations, expanding its horizons and perspective by including other domains and/or assessment tools that address various relevant aspects of life and interventions with OA. In this context, there are already assessment tools designed and/or validated in the OA population of Latin America and the Caribbean, for example, for elder abuse [40] and oral health in Mexico [41] and Colombia [42], health empowerment in Argentina [43], risk of falls at home in Cuba [44], and extrinsic fall risk assessment in residences in Chile [24]. Other tools have already been incorporated as part of public policy (for example, environment [25] and quality of life [24]). New tools could also be developed (for example, in areas such as quality of life, environment, health literacy, technology usability, user satisfaction with interventions, spirituality and religiosity, and legal, cultural, and ethnic aspects).

### 3.2. Comprehensive Gerontological Assessment Tools in Latin America and the Caribbean

With regards to the commonly used CGA tools in the countries of the region, a total of 50 have been compiled. Of these tools, 14 correspond to the biomedical or clinical domain (Table A1), which deals with health history, medication, protective and risk factors, health problems, diseases, and geriatric syndromes [4,26]. Sixteen tools belong to the mental domain (Table A2), of which eleven are cognitive tools that address baseline cognitive function, its alterations, and associated geriatric syndromes [4,26], and five are mood-related tools that focus on mood state and geriatric syndromes [4,26]. Twelve tools correspond to the functional domain (Table A3), which deals with baseline functional capacity, activities of daily living (basic, instrumental, and advanced), their impairments, disability and dependency, performance, and physical activity [4,26]. Eight tools correspond to the social domain (Table A4), which addresses the socio-familial network, participation, and socioeconomic aspects of OA and their environment [4,26]. Specifically, 28 tools have focused on identifying or categorizing geriatric syndromes.

The main countries where CGA tool validations have been identified, in order of frequency, are Chile (19), Colombia (14), Brazil (10), Mexico (8) and Argentina (6). Additionally, 14 validations that were conducted in Spain, which are used in the region due to the shared language, have been compiled (Table A1, Table A2, Table A3 and Table A4). Notably, the tools that show three or more validations in countries are Charlson comorbidity index (three validations), Mini Nutritional Assessment (five), Eating Assessment Tool-10 (three), MiniMental State Examination (five), Montreal Cognitive Assessment (six) and the Geriatric Depression Scale of Yessavage (three). Furthermore, the tools that have been used in government documents from three or more countries are the Mini Nutritional Assessment (four countries), the MiniMental State Examination (five), the Montreal Cognitive Assessment (three), the Geriatric Depression Scale of Yessavage (six), the Katz Index (seven), the Barthel Index (five), the Lawton–Brody Instrumental Activities of Daily Living Scale (four), Timed Up and Go (three), the Gijon Social-Family Scale (three) and the Zarit Caregiver Burden Interview (four) (Table A1, Table A2, Table A3 and Table A4).

It is important for doctors and professionals in the health and psychosocial fields, within their professional work at their service, center, or program with OA, to gain a deeper understanding of various aspects that can improve CGA implementation and management [5], either as a whole or for specific tools. This will help avoid a lack of knowledge, the associated risks, and biases in the interpretation of results [8,9].

To thoroughly understand the most commonly used CGA assessment tools in the region, these are presented by describing the tool’s name, objective, the dimensions it evaluates, the number of questions, cut-off scores and/or classification results, administration method, validity, and reliability (including the context and characteristics of the OA population analyzed, and the main psychometric analyses conducted), and how these tools have been included in the countries’ government documents (Table A1, Table A2, Table A3 and Table A4).

### 3.3. Comprehensive Gerontological Assessment During the COVID-19 Pandemic

The pandemic and the health measures implemented to manage it (quarantines, isolation, and restrictions on social mobility) imposed by health authorities in various countries worsened certain biopsychosocial health issues in OA, such as prevalent chronic diseases, geriatric syndromes, disability, and dependency. This also increased physical inactivity and mental health problems among OA [18], a situation that was described as a geriatric emergency [16]. In this context, the geriatrics team at a hospital in Colombia developed an abbreviated CGA proposal for emergency settings for OA with suspected or confirmed COVID-19, taking into account the CGA tools to be applied at the time of admission to guide the treatment plan [45].

At the same time, the pandemic affected the continuity of various interventions with this age group. However, it accelerated the implementation of telehealth for OA, their families, and caregivers, allowing for continued care and creating an opportunity for the development of CGA through telehealth [18,46,47]. It has been shown that CGA via telehealth was conducted using online questionnaires, videoconferencing, remote monitoring devices, mobile applications, among others, with a focus on evaluating the functional domain and activities of daily living [18]. Additionally, it has been documented that videoconference sessions were comparable to in-person visits in terms of cost, acceptance, and diagnostic accuracy [46], allowing for early disease detection, monitoring of chronic disease progression, providing personalized care, optimizing healthcare resources, and improving health outcomes for OA [18]. Furthermore, it has been reported that barriers for OA with visual or hearing disabilities were reduced [9]. Notably, some CGA proposals through telehealth have already been suggested in the United States [46] and Japan [18], that identify which domains can be evaluated through telehealth [18,46].

Among the CGA experiences through telehealth in the region, the Day Centers program for OA by the National Service for the Elderly (SENAMA) in Chile [48] gathered local experiences from some of its centers and convened a group of experts to design a centrally coordinated, multidimensional, abbreviated CGA. This CGA included key questions extracted from different CGA tools, forming a set of essential questions covering the various CGA domains (as a screening tool). It was administered via a phone interview [8] by any member of the team, allowing the exploration of the physical, functional, psycho-affective, and social aspects of OA. This enabled local teams to continue with specific evaluations and interventions through synchronous or asynchronous telehealth modality [48]. Other teams implemented these assessment tools through the Google Forms platform, allowing them to reach OA either by phone or email. This latter strategy has also been implemented in previous research and during the pandemic to optimize data management and validation in research [31].

## 4. Discussion

### 4.1. Challenges and Opportunities for Comprehensive Gerontological Assessment in Latin America and the Caribbean

#### 4.1.1. CGA Considerations from an Administration Perspective of Tools

A series of aspects should be considered when including/using an assessment tool as part of the CGA and as part of clinical reasoning in gerontology and geriatrics [8,9,27,39]; these aspects are detailed in Figure 2 and Table 1. Analyzing these factors will allow the professional team to decide which tool(s) are the best to include or use in a service, center, program, or intervention, and consequently achieve optimal analysis and interpretation of results.

The objective of the tool(s) or the CGA as a whole should be kept in mind, whether such an objective is clinical, where it is necessary to know the baseline state and the effect of the intervention on OA or whether it is for research, where it is essential to have tools validated in Spanish and/or Portuguese, to ensure the stability and reliability of their psychometric characteristics throughout the process [49]. It is important to highlight that, in several CGA tools, the goal is to screen for geriatric syndromes, where the tool’s questions arise from causal factors and/or symptoms that generate them (for example, the FRAIL scale: fatigue, resistance, ambulation, illnesses, loss of weight). Additionally, some of these same factors or symptoms (questions) are, in fact, other geriatric syndromes (cascade etiology); consequently, it is necessary to further evaluate the second geriatric syndrome that appeared as a factor or symptom of the first one, in order to understand the various needs of OA (for example, in the Mini Nutritional Assessment, where some altered questions reflect the presence of another geriatric syndrome; it is recommended to assess the second syndrome as well).

It will also be necessary to identify, in those tools that have two parts (e.g., the Goldberg Anxiety and Depression Scale or the Tinetti Gait and Balance Assessment Tool), whether these parts can be applied separately—only one part—or if they need to be administered together, and whether the scores from the different parts need to be combined. Additionally, it is important to identify tools that should be or are recommended to be applied together (e.g., One-Leg Balance and Timed Up and Go [24,50], or the Insomnia Severity Index and the Epworth Sleepiness Scale [24]), as they show better performance when used in combination or are complementary in analyzing the multiple causes and consequences of geriatric problems or syndromes [51].

Before administering the evaluation tool, it is also important that the person conducting the assessment and analyzing the results is familiar with the scoring direction and the overall score. In some tools, a higher score indicates a problem or deficit and vice versa (i.e., directly related, as with the Pfeffer Functional Activities Questionnaire, where a higher score indicates “functional impairment”), while in others, a higher score does not indicate a problem or deficit, and vice versa (i.e., inversely related, as with the Barthel Index, where a higher score indicates “independence”). Additionally, it is crucial to understand the minimum and maximum scores, the cut-off points (which are especially relevant in tools influenced by education level, such as cognitive tools), the classification provided by the tool, and/or the reference tables for performance in physical, functional, or cognitive clinical tests. Likewise, it is important to know whether the tool provides an overall result and/or results by dimensions, and conduct both analyses if applicable (e.g., the Food Quality Survey of Elderly, ECAAM). It is also necessary to delve into and specifically analyze questions/items that reflect altered results, revealing a particular problem or deficit (for example, the Montreal Cognitive Assessment, which, in addition to detecting cognitive impairment, identifies specific higher cognitive function alterations). All of this allows for a comprehensive view of the OA’s or caregiver’s results in the evaluated domain, enabling the identification of key areas for intervention.

It is essential to recognize the number and type of questions the assessment tool includes. A question with two response categories (e.g., Yes/No, as in the Yesavage abbreviated questionnaire) presents less difficulty for the professional to administer and is easier for the OA or their caregiver to understand than one with five categories (e.g., Never/Rarely/Sometimes/Most of the time/Always, as in the Medical Outcomes Study-Social Support Survey). Therefore, we propose some strategies to facilitate administration: ensure that the OA is using their glasses and/or hearing aids; detail and explain the purpose of the assessment and response options in advance; read the responses aloud in the case of low educational level (when there are few alternatives); laminate the response categories as a card and show this card while applying the tool, so that the OA can select the appropriate answer; provide the tool and read it together; and start from the response indicated by the OA, asking again to refine the answer toward the most accurate option (especially when there are more than 4–5 categories, as people often default to middle or extreme options).

It is important to consider the setting where the evaluation will take place, as some tools require a physical space for their execution (e.g., the Senior Fitness Test) or privacy and calm for their administration (e.g., the Yesavage abbreviated questionnaire or the Zarit caregiver burden scales). Additionally, in the case of clinical tests, it is necessary to ensure the safety conditions and availability of required equipment, as well as the relevance and feasibility of obtaining such equipment locally. It is also essential to consider the time needed to carry out the evaluation, which is directly related to the number of questions or response items included in the tool.

The CGA should be applied before and after interventions, ensuring that the application conditions are the same at the time of reevaluation. In the context of programs or services within public policy, it is recommended to assess regularly once a year [50,52], and consecutively each year throughout the lifespan of OA. However, in cases of frailty, this interval should be shorter, every 6 months, as it improves health outcomes [4]. Additionally, in situations such as serious illness, falls, hospitalization, or bed confinement, during and/or after these events, it is necessary to reevaluate the OA with the CGA [24]. It is also important to note in “observations” any relevant aspects associated with the assessment and/or adaptations that have been made, so that if the physician or professional conducting the reevaluation is not the same (due to leave, replacement, or a change in professional), the evaluation is conducted under the same initial conditions.

#### 4.1.2. CGA Considerations from an Academic Training Perspective

Topics related to CGA and its associated evaluation tools should be included as part of the regular curriculum in all undergraduate courses related to gerontology and geriatrics across various disciplines that address OA [3,8]. It is important to recognize that there is still a need for the regular inclusion of these subjects in the curricula of university-level professional careers. Similarly, it should be one of the core subjects within postgraduate programs (diplomas, master’s, and doctorates) for doctors and professionals in the region, considering the training gaps in the area [2,3], and aiming at promoting interdisciplinary work within the context of person-centered care [12,39].

It is also necessary to train and provide updated knowledge on CGA to professionals already working in various programs and services for OA [9], particularly in primary healthcare (PHC) [2,3]. Additionally, replicating micro-learning experiences, such as those already implemented in perioperative settings, has shown significant improvements in doctors’ management of CGA following this educational intervention [53]. As a strategy, it will be important for those beginning work with OA and CGA to self-administer the tool and/or apply it to a family member to become thoroughly familiar with it. Furthermore, self-training by watching application videos produced by the National Institute of Geriatrics in Mexico, which are available online [25,54], and taking specific courses on CGA would be beneficial.

#### 4.1.3. CGA Considerations from a Research Perspective

Efforts should be made to design, translate, culturally adapt, and/or determine the validity and reliability of assessment tools that address various aspects of the life of OA not yet covered by the CGA [9,27]. Additionally, validations of the same tool in two or more countries, as done in Argentina and Chile [55], should be carried out, incorporating all these tools into the regular use of CGA in the region. This presents an opportunity to transfer scales, questionnaires, or clinical tests from CGA to be used in telehealth evaluations of OA and their caregivers [18]. It may even be possible to create a repository of validated CGA tools in the region under the COMLAT-IAGG (Latin American and Caribbean Committee—International Association of Gerontology and Geriatrics) as a regional branch or under IAGG globally, following the good practices implemented in the field of psychology in Argentina by the Society for Psychotherapy Research [56]. Alternatively, it would also be beneficial to create websites with psychometric information and/or administration details about the assessment tools, as it was done with The Fototest in Spain [57].

In parallel, it is essential to advance qualitative, quantitative, and mixed research with an interdisciplinary emphasis from gerontology, recognizing the particularities and strengths of our region. This research should aim to diagnose how well medical, professionals and healthcare providers know and utilize the CGA tools [58], or which CGA tools use to address geriatric syndromes [59], similar to studies that have already been conducted in geriatricians in European countries. Furthermore, efforts should be made to apply CGA tools at the Latin American, national, local, and multicentric levels, targeting large population groups or cohorts of OA, as this is not regularly addressed in national public health surveys in the countries. This will enable the validation of CGA tools, as it has been done in Mexico [60].

Such initiatives will help generate new databases, allowing for a more precise analysis and understanding of OA and their intervention needs [27], with a special emphasis on groups of greater interest, where evidence is still limited, such as octogenarians, nonagenarians, and centenarians in the region. All of this will facilitate the development of a research line at the regional level. Additionally, updating and consolidating the validated tools in the region is crucial, with the current compilation (Table A1, Table A2, Table A3 and Table A4) serving as a valuable resource for the methodological aspects of undergraduate and postgraduate theses, as well as for research projects.

#### 4.1.4. CGA Considerations from an Innovation Perspective

The COVID-19 pandemic left a series of lessons regarding CGA implementation through telehealth [18,46,47] which can be adapted to the realities of the region. Therefore, efforts should be made to advance innovation and research aimed at determining the validity and reliability of CGA tools when administered via telephone or video, comparing these methods with in-person assessments, as has already been done with the Barthel Index conducted over the phone [21].

Given that telehealth experiences have diminished following the return to in-person services in the countries of the region, it is necessary to revisit them and generate new ones that can serve as a regular strategy for doctors and professionals working with OA [18]. These strategies are not only useful in a pandemic context, but also in situations of natural disasters, rural settings, disability, and caregiving, which limit social mobility and/or restrict the socio-health participation of OA.

In addition, web applications can be designed and implemented that include the CGA and/or its evaluation tools, such as the Dependency Indices or GeriatriApp [61] developed in Spanish, GeriKit in English [62], or ICOPE (Integrated Care for Older People) in both languages [63]. These applications can include visual and/or auditory elements that also allow for long-term follow-up of OA with cognitive impairments [9], which presents a challenge for both innovation and research.

#### 4.1.5. CGA Considerations from a Management Perspective

Proposals should be advanced for a limited screening of the CGA, following previous experiences [48,64], which would allow for a more comprehensive CGA to be conducted later [10]. This includes developing batteries of CGA tools to be used at different levels of care and in social health programs as part of public policy in the countries of the region [2]. Such initiatives have already been implemented in primary care settings [50] or in residences for OA in Chile [24], thus creating a contextually relevant CGA application aligned with social healthcare models and levels of attention [39]. This offers the possibility of generating new contextually tailored CGA proposals for day centers for OA and community rehabilitation centers; in hospital settings such as day hospitals, emergency rooms, general inpatient wards, outpatient clinics [37], and home care [65], as documented in Colombia; and specialized geriatric units within hospitals [6,9], such as orthogeriatric units in Colombia [66] and acute geriatric units in Chile [67]. This approach facilitates the continuity and sustainability of interventions for OA across various levels of social healthcare in a coordinated manner, which we propose in Figure 3 and Table 2, based on this literature review and the authors’ experiences.

Similarly, it is important to advance contextualized CGA that address major specified issues such as dementia, disability, oncology, or palliative care [8,70]. At the same time, CGA units could be established within the array of programs and services for OA, as is the case in Uruguay [2,34]. Additionally, in certain programs or centers, CGA results could be integrated with intervention protocols for managing OA, preventing, and addressing geriatric syndromes—a practice already implemented in nursing homes as part of public policy in Chile, stemming from collaborative academic, expert, and interdisciplinary efforts [24]. Furthermore, these assessments should be linked to multicomponent interventions. The results from applying CGA tools should also be set as management goals within public policy, as seen in Panama [52].

It is expected that the Interuniversity Network for Healthy Aging in Latin America and the Caribbean (RIES-LAC) [71] will serve as a platform for generating collaborative networks, research, community engagement, advanced human capital training, and contributions to public policies. Drawing on the expertise of academics from across the region, it will enhance CGA, considering that the four micro networks within this network, from both gerontology and geriatrics, used the four traditional spheres of CGA in their conceptual definitions, thereby contributing to the CGA as a whole [71].

One of the limitations of this study is that it did not assess which CGA tool is most commonly used in each country; it identified instead which tools are used or recommended in each country based on government documents. Furthermore, no government documents were found concerning the utilization of CGA tools in countries with an incipient demographic transition and low rates of population aging. Despite these limitations, our study updated the concept of Comprehensive Gerontological Assessment for its application in the Latin America and Caribbean region, which can serve as a reference for other regions of the world. The compilation of the different CGA tools, the examination of CGA adaptations during the COVID-19 pandemic, and the challenges related to CGA administration, training, research, innovation, and management can provide useful information for health practitioners and policy makers, that can contribute to the subsequent elaboration of more effective, multidimensional care plans for OA.

## 5. Conclusions

Comprehensive Gerontological Assessment (CGA) is an essential tool for addressing the needs of OA; however, there is a need for greater knowledge and training on its various assessment tools. This literature review highlights the importance of evaluation in the context of gerontology, while also synthesizing the validated and most commonly used tools in Latin America and the Caribbean region. It proposes considerations for the inclusion of these tools and outlines CGA battery tools for different levels of care, programs, and socio-health services for OA. This is of particular interest as it offers lines of action for decision-makers, academics, researchers, university students, and the community at large. Consequently, the Comprehensive Gerontological Assessment (CGA) batteries presented in this literature review establish a minimum standard for implementation in care models and public policies for the countries of Latin America and the Caribbean.

We recommend that the professionals working with OA become familiar with the different spheres of CGA and its evaluation tools, by practicing and applying them regularly, so as to develop greater skills in their application, analysis, and interpretation. This will enable more tailored interventions that meet the needs of OA, their families, and caregivers, promoting their rights, which will lead to improved functionality, autonomy, and quality of life, while also fostering healthy, active, and successful aging.

## Figures and Tables

**Figure 1 ijerph-21-01697-f001:**
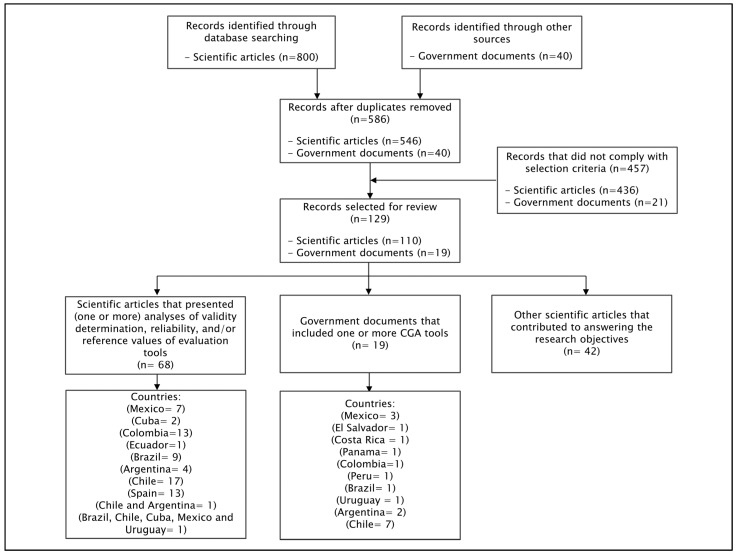
Diagram of the article selection process for the literature review, categorized by type of document and country. Abbreviation: CGA: Comprehensive Gerontological Assessment.

**Figure 2 ijerph-21-01697-f002:**
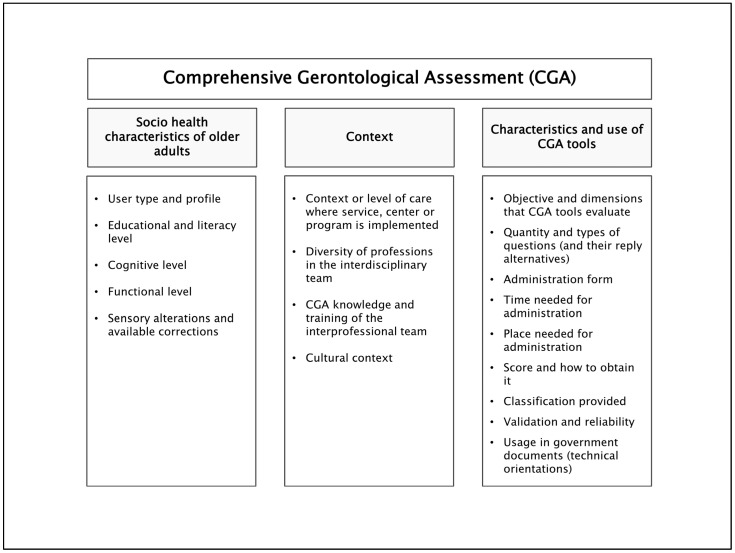
Summary of key aspects to consider as part of clinical reasoning by the interdisciplinary team at the moment of including/using an assessment tool as part of the CGA. Abbreviation: CGA: Comprehensive Gerontological Assessment.

**Figure 3 ijerph-21-01697-f003:**
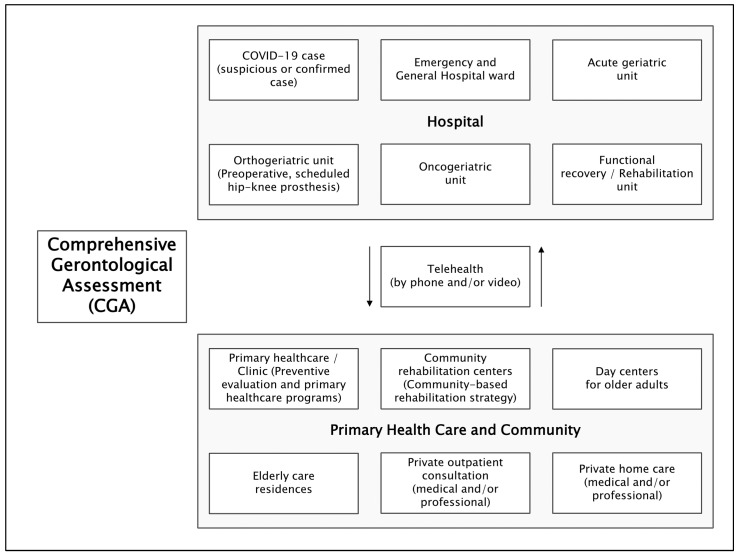
Levels of care, programs, and social-health services for older adults in Latin America and the Caribbean, where a Comprehensive Gerontological Assessment battery is proposed. Abbreviation: CGA: Comprehensive Gerontological Assessment.

**Table 1 ijerph-21-01697-t001:** Aspects to consider as part of clinical reasoning by the interdisciplinary team at the moment of including/using an assessment tool as part of the CGA.

Aspects	Considerations (Identify/Revise)
OA user type and profile	- Type of family: OA living alone, OA living with a partner, OA with a nuclear family, OA with an extended family, OA with/without a caregiver (formal or informal). - Economic resources that allow them to travel to the evaluation site. - Presence of geriatric syndromes and/or the need for identification and/or severity assessment. - Personal characteristics: education level, cognitive, functional, and sensory abilities.
Educational and literacy level of OA	- OA who are illiterate, with primary, secondary, or technical/university education. - Ability to read and write independently or with assistance.
Cognitive level of OA	- OA with normal cognitive state, mild, moderate, or severe cognitive impairment. - Impairment of one or more higher cognitive functions. - Whether they are able to answer the questions asked during the evaluation or if they require assistance.
Functional level of OA	- OA who are independent, with mild, moderate, or severe dependence. - Whether they walk (alone or with assistance), remain seated, or are bedridden (and their level of mobility). - Requirement for assistive mobility and transfer devices (cane, walker, or wheelchair).
Sensory alterations of OA and available corrections	- Visual or auditory impairments that OA may present. - Correction of impairments, with glasses or hearing aids.
Context or level of care where service, center or program is implemented	- General intervention models in gerontology and geriatrics (social, mixed, or health). - Levels of health care and their emphasis (primary: promotional and preventive; secondary: diagnosis and treatment; tertiary: care, intervention, and specialized rehabilitation). - Specific programs and services for OA (Emergency and General Hospitalization Room, Acute Geriatric Unit, Orthogeriatric Unit, Oncogeriatric Unit, Functional Recovery/Rehabilitation Unit, Day Hospital, Memory Unit, Incontinence Unit, primary health care or clinic and specific programs for OA at this level, community rehabilitation center (in PHC), day centers for OA, community programs, residential care for OA, private outpatient care, home care, among others).
Diversity of professions in the interdisciplinary team	- Presence or absence. - Wide or limited. - Presence and diversity of potential professionals who can apply and analyze the tool(s) (geriatrician, primary health care physician, kinesiologist or physiotherapist, speech therapist, occupational therapist, nutritionist, nurse, midwife, dentist, pharmacist, psychologist, social worker or social assistant, gerontologist, physical education teacher, psychometrician, music therapist, and other professionals working with OA).
CGA knowledge and training of the interprofessional team	- Knowledge and familiarity with the tools. - Number of interdisciplinary team professionals trained in the CGA tool(s) (one or several). - Post-training needs.
Cultural context	- Urban or rural. - Particular geographical areas (coast, valley, mountain range). - Particular ethnic contexts.
Objective and dimensions that CGA tools evaluate	- Main objective of tool. - Secondary objectives. - Characteristics of each question (and whether they address cause, symptom, condition, or another geriatric syndrome).
Quantity and types of questions (and their alternative replies)	- Administration time. - Complexity of understanding by OA. - Requirement for detailed explanation by professional. - Complexity of administration by professional.
Administration form	- Considering different administration alternatives: self-administered, administered by professional to OA, administered by professional to caregiver or family member, observation, administration by phone, administration via telehealth. Some tools allow only one form of administration, while others allow multiple forms.
Time needed for administration	- Time in minutes, identifying time required for its completion, whether within a single session (part of the session or the entire session) or over multiple evaluation sessions.
Place needed for administration	- Location, identifying space required and necessary implements for its completion. - Conditions of privacy and tranquility for OA and/or their caregiver.
Score and how to obtain it	- Tool allows for a score to be obtained. - Method and/or calculation required to obtain the score (easy or difficult). - Cutoff score based on validation. - This score allows for classification/categorization. - Levels of classification or categorization produced by score. - It is recommended that cutoff points be defined based on formal education.
Classification provided	- Tool is for screening, diagnosis, or clinical analysis. - Feasibility of generating severity categories for analyzed condition. - Feasibility of guiding intervention with these severity categories.
Validation and reliability	- Construct, content, or criterion validity (concurrent or predictive) of tool, in the language or country (or region). - Reliability according to evaluation tool, time, and/or observer; method used and analysis of its results.
Usage in the government documents (technical orientations)	- Most commonly used and recommended. - Forms and administration recommendations used/provided.

Abbreviations: CGA: Comprehensive Gerontological Assessment; OA: older adults; PHC: primary healthcare.

**Table 2 ijerph-21-01697-t002:** Compilation and proposals for Comprehensive Gerontological Assessment batteries for different levels of care, programs, and social health services for older adults in the context of Latin America and the Caribbean. Adapted from publications and technical guidelines of countries in the region, and based on authors’ experience.

Level of Care, Service or Program	Biomedical	Mental (Cognitive and Emotional)	Functional	Social
COVID-19 case (suspicious or confirmed case) Adapted from [45]	- Assessment of index pathology - CCI - Assessment of comorbidity and risk factors: Hypertension Ischemic heart disease Heart failure COPD Asthma Type 2 diabetes Immunosuppression Chronic kidney disease Liver disease Obesity Malnutrition - Requested paraclinical tests: - ETADI	- CAM - Investigate previous diagnosis of dementia or major neurocognitive disorder. - Subjective complaint of memory or cognitive decline - MMSE (if possible and in case of doubt regarding cognitive diagnosis)	- Barthel Index - IADL - FRAIL only if Barthel Index > 90	- The Gijon social-family scale - Semi-structured interview: Housing conditions Room conditions Number of bathrooms in house Presence of caregiver
Emergency and General Hospital Ward Adapted from [37]	- CCI - EAT-10 - MNA-SF - Norton Scale - Downton fall risk assessment scale - Falls within previous year. - ETADI	- CAM - MMSE - MoCa - SPMSQ - Subjective memory complaint - GDS - NPI, if applicable	- Barthel Index - IADL - FRAIL - SARC-F	- LSNS-6 - Ecomap
Acute Geriatric Unit Adapted from [67]	- CCI - EAT-10 - MNA-SF - Norton Scale - Downton fall risk assessment scale. - ETADI	- CAM - MMSE - MoCA - SPMSQ - GDS - NPI, if applicable	- Barthel Index - IADL - TUG - Tinetti Gait and Balance Assessment Tool - PFAQ	- Genogram - Ecomap - LSNS-6 - ZCBI
Orthogeriatric Unit (Preoperative, scheduled hip-knee prosthesis)	- CCI - EAT-10 - MNA-SF - Norton Scale - Downton fall risk assessment scale - ETADI	- CAM - MMSE or SPMSQ (in case of low educational level and/or rurality) - GDS - GADS (Anxiety Scale) - Short FES-I	- TFI - Barthel Index - IADL - Tinetti Gait and Balance Assessment Tool - SPPB	- Genogram - Ecomap - ZCBI - EQ-5D + EQ-VAS
Oncogeriatric Unit Adapted from [68,69]	G8 (oncological) (a) G8 ≤ 14 points, coordinated for CGA (within no more than 7 days) (b) G8 > 14 points, continuation with regular care in oncology			
	- CCI - MNA-SF - Downton fall risk assessment scale - ETADI - Risk of chemotherapy toxicity (Chemo-Toxicity Calculator)	- MMSE - MoCA - SPMSQ - GDS - GADS (Anxiety Scale)	- Barthel Index - IADL - Tinetti Gait and Balance Assessment Tool - SPPB	- The Gijon social-family scale - MOS-SSS - ZCBI or abbreviated ZCBI - EQ-5D + EQ-VAS
Functional Recovery/ Rehabilitation Unit Adapted from [37]	- CCI - EAT-10 - MNA - Norton Scale - Downton fall risk assessment scale - Falls within previous year - ETADI - ICIQ-SF	- CAM - MMSE - MoCa - Subjective memory complaint - GDS - GADS (Anxiety Scale) - FES-I - NPI, if applicable	- Barthel Index - IADL - TADL-Q - FRAIL - TFI - SARC-F - Dynamometry - PFAQ - Tinetti Gait and Balance Assessment Tool - BESTest - SPPB	- Genogram - Ecomap - ZCBI - MOS-SSS - EQ-5D + EQ-VAS
Primary Health Care/Clinic (Preventive Evaluation and Primary Health Care Programs) Adapted from [50]	- ICIQ-SF - Downton fall risk assessment scale - ECAAM	- MMSE - ISI + ESS - GDS - GADS	- Barthel Index - IADL - TFI - SARC-F - PFAQ - EVA - One-Leg Balance test + TUG - SFT (Autonomous and Slightly Dependent) - SPPB (Fragile, Moderately Dependent, and Disability)	- Gerogram - Ecomap - MOS-SSS - EQ-5D + EQ-VAS
Community Rehabilitation Centers (Community-Based Rehabilitation Strategy)	- ETADI - ICIQ-SF - MNA - ECAAM - EAT-10 - Downton fall risk assessment scale - Norton Scale (only if necessary)	- MMSE o Pfeiffer (in case of low educational level and/or rurality) - MoCA - ISI + ESS - GDS - GADS - FES-I	- Barthel Index - IADL - TADL-Q - One-Leg Balance test + TUG - Tinetti Gait and Balance Assessment Tool - BESTest - EVA - SFT (Autonomous and Slightly Dependent) - SPPB (Fragile, Moderately Dependent, and Disability)	- Genogram - Ecomap - MOS-SSS - ZCBI - EQ-5D + EQ-VAS
Day Centers for Older Adults Adapted from [48]	First 48 h: - Admission form - MNA - Norton Scale (only if necessary)	First 48 h: - CAM (only in case it is necessary) - MMSE or SPMSQ (in case of low educational level and/or rurality)	First 48 h: - Barthel Index - TFI - One-Leg Balance test + TUG - EVA	First 48 h: - Personal data and contact
	Complete until first week: - ICIQ-SF - ETADI - EAT-10 - ECAAM	Complete until first week: - ISI + ESS - GDS - GADS (Anxiety Scale) - Short FES-I	Complete until first week: - IADL - Tinetti Gait and Balance Assessment Tool - SFT (Autonomous and Slightly Dependent) - SPPB (Fragile, Moderately Dependent, and Disability)	Complete until first week: - Ecomap - MOS-SSS - ZCBI - EQ-5D + EQ-VAS
Elderly Care Residences Adapted from [24]	First 24 h: - Admission form - Downton fall risk assessment scale - Norton Scale	First 24 h: - CAM - MMSE or SPMSQ (in case of low educational level and/or rurality) - Inquiry about mood aspects in the admission form	First 24 h: - Barthel Index - PFAQ - EVA	First 24 h: - Update of social background.
	Complete until second week: - ETADI - ICIQ-SF - MNA - EAT-10	Complete until second week: - ISI + ESS - GDS - GADS (Anxiety Scale) - NPI, if applicable	Complete until second week: - IADL - TADL-Q - Six-Minute Walk Test - RPE - One-Leg Balance test + TUG - Tinetti Gait and Balance Assessment Tool - BESTest - SFT (Autonomous and Slightly Dependent) - SPPB (Fragile, Moderately Dependent, and Disability)	Complete until second week: - Genogram - Ecomap - MOS-SSS - EQ-5D + EQ-VAS
Private outpatient consultation (medical and/or professional) Adapted from [37]	- EAT-10 - MNA-SF - Downton fall risk assessment scale - Falls within previous year - ETADI	- CAM, if applicable - MMSE - MoCA - Subjective memory complaint - GDS - FES-I - GADS (Anxiety Scale) - NPI, if applicable	- Barthel Index - IADL - FRAIL - SARC-F - SPPB	- Genogram - Ecomap - ZCBI - MOS-SSS
Private home care (medical and/or professional) Adapted from [37]	- EAT-10 - MNA-SF - ECAAM - Norton Scale - Downton fall risk assessment scale - ETADI	- CAM - MMSE - MoCa - GDS - FES-I - GADS (Anxiety Scale) - NPI, if applicable	- Barthel Index - IADL - TADL-Q - FRAIL - SARC-F - One-Leg Balance test + TUG - Tinetti Gait and Balance Assessment Tool - SPPB	- Genogram - Ecomap - ZCBI - MOS-SSS
Telehealth (by phone and/or video) Adapted from [18,46]	- MNA-SF - ECAAM - EAT-10 - Downton fall risk assessment scale - ETADI (by video)	- CAM - MMSE - SPMSQ - MoCA - GDS - GADS (Anxiety Scale)	- Barthel Index - IADL - Standing up and sitting down from the chair, gait, TUG within the living space (via video) - EVA - TADL-Q	- MOS-SSS - ZCBI

Abbreviations: CCI: Charlson comorbidity index; COPD: Chronic Obstructive Pulmonary Disease; EAT-10: Eating Assessment Tool-10; MNA: Mini-Nutritional Assessment; MNA-SF: Mini-Nutritional Assessment-Short Form; ETADI: Denmark dismobility stage; CAM: Confusion Assessment Method; MMSE: Mini-Mental State Examination; MoCa: Montreal Cognitive Assessment; SPMSQ: Short Portable Mental Status Questionnaire of Pfeiffer; GDS: The Geriatric Depression Scale of Yessavage; IADL: Lawton–Brody Instrumental Activities of Daily Living Scale; TUG: Timed up and go; PFAQ: Pfeffer Functional Activities Questionnaire; GADS: Goldberg Anxiety and Depression Scale; FES-I: Falls Efficacy Scale-International; Short FES-I: The short Falls Efficacy Scale-International; TFI: Tilburg Frailty Indicator; SPPB: Short Physical Performance Battery; ZCBI: Zarit Caregiver Burden Interview; NPI: Neuropsychiatric Inventory; LSNS-6: The Lubben-6 Social Network Scale; G8: Geriatric 8; FRAIL: FRAIL Scale (Fatigue, Resistance, Ambulation, Illnesses, Loss of Weight); SARC-F: SARC-F (Strength, Assistance in walking, Rise from chair, Climb stairs—Falls) questionnaire; ICIQ-SF: International Consultation on Incontinence Questionnaire Short-Form; ECAAM: Food Quality Survey of Elderly; ISI: Insomnia Severity Index; ESS: Epworth Sleepiness Scale; MOS-SSS: The Medical Outcome Study Social Support Survey; SFT: Senior Fitness Test; TADL-Q: The Technology—Activities of Daily Living Questionnaire; BESTest: Balance Evaluation Systems Test; Q-5D: EuroQol-five dimensions; EQ-VAS: EuroQol-visual analogue scale; EVA: Visual Analogue Scale; RPE: Borg Rating of Perceived Exertion Scale.

## Data Availability

Not applicable.

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
