# Peer review of "Comprehensive Gerontological Assessment: An Update on the Concept and Its Evaluation Tools in Latin America and the Caribbean—A Literature Review"

_ijerph, 2024, doi:10.3390/ijerph21121697_

Round 1

Reviewer 1 Report

Comments and Suggestions for Authors

The abstract of the study is quite well structured and explains in a concise and effective way the aim of the paper which is to offer a systematic review of the methodologies and tools used in the CGA of the Latin American area.

Even if the approach of the study is well set up, some problems are noted in the research methodology.As regards the research methodology of the literature and studies conducted on the topic, there is no explanatory table that explains the methods of selecting the studies and the hierarchy of their evaluation. The exclusion and inclusion criteria of the studies should therefore be better described also highlighting a reference methodology. Tailored search strategies, combining free text keywords and vocabulary terms, must be reported in detail in a Table  of Supplemental Materials, should be developed by an information specialist (MDC) and further refined through team discussion. Reference lists of retrieved articles must be reviewed for further evidence. The final search results should  be  exported into EndNote, and duplicates should be removed. 

As for minor aspects, I would arrange Table 1 ( aspects to consider as part of clinical reasoning by the interdisciplinary team at the moment 330 of including / using an assessment tool as part of the CGA) in a more graphically acceptable format. For example, it could be useful to structure an infographic that allows for a more effective visualization of the highlighted key points.

The same reasoning and considerations are applicable to Table 2 (Compilation and proposals for Comprehensive Gerontological Assessment batteries for 500 different levels of care, programs, and social health services for older adults in the context of Latin 501 America and the Caribbean. Adapted from publications and technical guidelines of countries in the 502 region, and based on authors' experience).

For the rest I do not find any other potentially critical elements. the paper is well written and offers a good summary of all the material cited in the search strategy.

Furthermore, the conclusions should be expanded by defining, based on what has been highlighted by the studies and tools used in the Latin American area, which are the most effective, rapid and easy-to-use ones to propose to stakeholders to organize services for the elderly both in hospital settings and in the context of home care or in the context of long-term care.

Author Response

Response to Reviewer 1

Comment:

The abstract of the study is quite well structured and explains in a concise and effective way the aim of the paper which is to offer a systematic review of the methodologies and tools used in the CGA of the Latin American area.

Response: Thank you.

Comment:

Even if the approach of the study is well set up, some problems are noted in the research methodology.As regards the research methodology of the literature and studies conducted on the topic, there is no explanatory table that explains the methods of selecting the studies and the hierarchy of their evaluation. The exclusion and inclusion criteria of the studies should therefore be better described also highlighting a reference methodology. Tailored search strategies, combining free text keywords and vocabulary terms, must be reported in detail in a Table  of Supplemental Materials, should be developed by an information specialist (MDC) and further refined through team discussion. Reference lists of retrieved articles must be reviewed for further evidence. The final search results should be exported into EndNote, and duplicates should be removed.

Response: Thank you very much for your comment. We would like to underline that, as stated in the title, abstract, research objective, and methodology, the presented article is a "literature review." To support this aspect, we have referenced the scientific article titled "A typology of reviews: an analysis of 14 review types and associated methodologies." This article explains that literature reviews can cover a wide range of topics with varying levels of completeness and thoroughness, based on the analysis of the literature. These reviews may include research findings; the search method may or may not involve an exhaustive search; they may or may not include quality assessment; their synthesis is typically narrative; and the analysis can be chronological, conceptual, thematic, etc.

Therefore, we believe that to meet the objectives of this research, the design that best fits is a literature review. Below, we recall the objectives of this research (highlighted in red in the text):

“a) conceptualize Comprehensive Gerontological Assessment (CGA) for its use in the Latin America and Caribbean region; b) analyze the characteristics of the validated and most commonly-used CGA tools in the region; c) analyze the adaptations made to CGA during the COVID-19 pandemic in the region; d) discuss challenges and opportunities regarding administration, academic training, research, innovation, and management of CGA in the region, which will lead to the improvement of CGA management by doctors and professionals, and enhance the quality of life for OA.”

This is also reflected in the methodology with the phrase (highlighted in red in the text):

“A literature review was chosen as it allows for a broader and more diverse identification of available literature, as well as a more flexible and contextual exploration [22]. To achieve the aims of this review [22], a literature search was conducted to answer the question: How was the Comprehensive Gerontological Assessment (CGA) and its assessment tools implemented in Latin America and the Caribbean before, during, and after the COVID-19 pandemic?”

[22] Grant M, Booth A. A typology of reviews: an analysis of 14 review types and associated methodologies. Health Info Libr J 2009;26:91–108. https://doi.org/10.1111/J.1471-1842.2009.00848.X.

Following the reviewer's recommendations, the inclusion and exclusion criteria, as well as the steps taken to conduct this literature review and its respective bibliographic reference, have been added to the methodology section of the article with the following phrase (highlighted in blue in the text):

“Scientific articles were included if they contributed to answering the objectives of the literature review in conceptual and contextual terms or if they determined the validity, reliability, and/or provided reference values for any of the assessment tools in the countries of Latin America and the Caribbean. This could involve studies focusing exclusively on populations aged 60 years and older or as part of the general population.

Additionally, government documents and technical guidelines from countries in the region were included, whether they focused specifically on older populations or the general population. These documents needed to include any of the tools identified as valid and/or reliable in this review or propose CGA batteries for older adults at different levels of care, programs, and/or social-health services for older adults in countries in the Latin America and Caribbean region. 

Abstracts from conferences, scientific congress reports, dissertations, research re-ports, theses, letters to the editor, editorials, and commentaries were excluded.

The following steps were followed in the literature review process [23] depending on the type of document. In the case of scientific articles, the steps were: strategy-based search, article identification, duplicate elimination, title and abstract screening, full-text reading, selection, compilation into tables, analysis, interpretation of results and discussion, and referencing. Using the list of assessment tools for which validity, reliability, and/or reference values were determined, the review process proceeded to government documents from countries in the region. In the case of government documents, the steps were: search, identification, full-text reading focused on identifying the selected tools and CGA battery proposals mentioned in the document, selection, compilation into tables, analysis, interpretation of results and discussion, and referencing. A reference librarian, part of the research team, collaborated throughout the various stages of the literature review process.

A total of 800 scientific articles and 40 government documents were initially identified for the literature review. There were 254 duplicates, which were eliminated. After applying inclusion and exclusion criteria, 129 documents were finally selected for the literature review (110 scientific articles and 19 government documents). Figure 1 provides a diagram of the article selection process for the literature review.”

Along with this, bibliographic reference number 23 has been added to support the steps followed in the literature review:

[23] Chigbu U, Atiku S, Du Plessis C. The Science of Literature Reviews: Searching, Identifying, Selecting, and Synthesising. Publications 2023;11:2. https://doi.org/10.3390/PUBLICATIONS11010002.

We have also added a figure (Figure 1: Diagram of the article selection process for the literature review, categorized by type of document and country; highlighted in blue in the text) that illustrates the flow of eligible, excluded, and selected articles and documents for the literature review. These are further categorized by quantity, type of document, and country of origin, directly relating to the bibliographic references of the submitted scientific article.

Comment:

As for minor aspects, I would arrange Table 1 (aspects to consider as part of clinical reasoning by the interdisciplinary team at the moment 330 of including / using an assessment tool as part of the CGA) in a more graphically acceptable format. For example, it could be useful to structure an infographic that allows for a more effective visualization of the highlighted key points.

Response: Thank you. We have created a figure that provides a general summary of the key aspects that should be reviewed and/or considered as part of the clinical reasoning process by the professional or interdisciplinary team when including/using an evaluation tool as part of the CGA. This figure offers a quicker and more precise overview, serving as a prelude to exploring Table 1 in greater depth. It is the new Figure 2: Summary of key aspects to consider as part of clinical reasoning by the interdisciplinary team at the moment of including / using an assessment tool as part of the CGA (highlighted in blue in the text).

Comment:

The same reasoning and considerations are applicable to Table 2 (Compilation and proposals for Comprehensive Gerontological Assessment batteries for 500 different levels of care, programs, and social health services for older adults in the context of Latin 501 America and the Caribbean. Adapted from publications and technical guidelines of countries in the 502 region, and based on authors' experience).

Response: Thank you. We have created a figure that provides a general summary of the different levels of care, programs, and social-health services for older adults in the context of Latin America and the Caribbean, which serve as the basis for proposing CGA batteries derived from this literature review. This figure offers a quicker and more precise overview, serving as a prelude to exploring Table 2 in greater depth. It is the new Figure 3: Levels of care, programs, and social-health services for older adults in Latin America and the Caribbean, where a Comprehensive Gerontological Assessment battery is proposed (highlighted in blue in the text).

Comment:

For the rest I do not find any other potentially critical elements. the paper is well written and offers a good summary of all the material cited in the search strategy.

Response: Thank you very much for your comment.

Comment:

Furthermore, the conclusions should be expanded by defining, based on what has been highlighted by the studies and tools used in the Latin American area, which are the most effective, rapid and easy-to-use ones to propose to stakeholders to organize services for the elderly both in hospital settings and in the context of home care or in the context of long-term care.

Response: Thank you. We have added the following phrase to the conclusion to highlight the importance of the proposed CGA batteries and their respective evaluation tools for different levels of care, programs, and social-health services for older adults in the context of Latin America and the Caribbean (highlighted in blue in the text):

"Consequently, the Comprehensive Gerontological Assessment (CGA) batteries presented in this literature review establish a minimum standard for implementation in care models and public policies for the countries of Latin America and the Caribbean."

Thank you very much for your valuable comments, which allow us to deliver an even more strengthened scientific article for publication in the journal.

Reviewer 2 Report

Comments and Suggestions for Authors

Dear authors

Thank you for the opportunity to review your paper on 'Comprehensive gerontological assessment: An update on the concept and its evalution tools in Latin America and The Carribbean: A literature review'.

The content of the paper is of a contemporary nature and could surely be used fruitfully by practitioners in the field of geriatrics and gerontology in other developing countries.

The paper is detailed and valuable. However, I have two major concerns:

1) The paper is in essence not a literature review, but a quasi-scoping review. You refer to databases consulted and search strings used. Unfortunately the rest of the paper fails to show the rigour of a scoping review. It is therefore strongly recommended that you revise that literature review, and add the phases necessary to meet the requirements of a scoping review (please do not confuse the recommendation with a systematic review). The value and scientific rigour of the paper could be greatly improved but adopt scoping review methodology. Apart from the detailed list of assessment instruments available, you will thus also provide the reader with a data chart referring to all the literature informing the review.

It goes without saying that your 'materials and methods' section will have to be much more detailed to meet the requirements of a scoping review, and you will have to include a PRIMSA-Scoping review.

2) With the inclusion of COVID-19 you bring a whole new dimension to the paper. I would strongly recommend that you separate the issues. Namely, you can do a separate paper for another journal on the gerontological assessment practices that were necessary during COVID-19. Keep the focus of the present paper purely on the assessment practices during 'normal' circumstances.

In terms of the layout the following:

1) It was noted that you often critique assessment instruments and practices while you are busy outlining the results of the study. Please keep the results section focused on what the literature indicates.

2) Include a section after the results called 'Discussion' - here you can debate the assessment tools and include critique where warranted.

3) In the conclusion section you can add 'recommendations' for future research, practice, training, continiuning professional development, policy, etc. 

It is not helpful to 'mix' all these 'key ingredients' of a paper under the one heading related to 'Results'.

Also see attached the manuscript with some additional comments.

I trust these comments will be helpful to improve your paper.

Comments on the Quality of English Language

Dear authors

The entire manuscript will have to be language edited. In some instances one sentences is the length of an entire paragraph. A reader looses comprehension when sentences are extra long. The writing must be clear, focused and word economic.

Author Response

Response to Reviewer 2

Comment:

Dear authors

Thank you for the opportunity to review your paper on 'Comprehensive gerontological assessment: An update on the concept and its evalution tools in Latin America and The Carribbean: A literature review'.

The content of the paper is of a contemporary nature and could surely be used fruitfully by practitioners in the field of geriatrics and gerontology in other developing countries.

Response: Thank you.

Comment:

The paper is detailed and valuable. However, I have two major concerns:

1) The paper is in essence not a literature review, but a quasi-scoping review. You refer to databases consulted and search strings used. Unfortunately the rest of the paper fails to show the rigour of a scoping review. It is therefore strongly recommended that you revise that literature review, and add the phases necessary to meet the requirements of a scoping review (please do not confuse the recommendation with a systematic review). The value and scientific rigour of the paper could be greatly improved but adopt scoping review methodology. Apart from the detailed list of assessment instruments available, you will thus also provide the reader with a data chart referring to all the literature informing the review.

It goes without saying that your 'materials and methods' section will have to be much more detailed to meet the requirements of a scoping review, and you will have to include a PRIMSA-Scoping review.

Response: We thank the reviewer for their thorough review and constructive comments regarding the methodology and rigor of our literature review. We value the recommendation to consider scoping review methodology as a way to enhance the scientific rigor of the article.

The objective of this article is to provide a narrative synthesis of the existing literature on Comprehensive Gerontological Assessment (CGA). Therefore, we believe that the design that best fits the objectives of this research is a "literature review." Below, we reiterate the objectives of this research (highlighted in red in the text):

“a) conceptualize Comprehensive Gerontological Assessment for its use in the Latin America and Caribbean region; b) analyze the characteristics of the validated and most commonly-used CGA tools in the region; c) analyze the adaptations made to CGA during the COVID-19 pandemic in the region; d) discuss challenges and opportunities regarding administration, academic training, research, innovation, and management of CGA in the region, which will lead to the improvement of CGA management by doctors and professionals, and enhance the quality of life for OA.”

Unlike scoping reviews, which aim to map the breadth and depth of evidence on a topic and identify research gaps, our literature review focuses on synthesizing key findings to address specific research questions relevant to the use and characteristics of CGA tools in Latin America and the Caribbean.

To support this aspect, we have referenced the scientific article titled "A typology of reviews: an analysis of 14 review types and associated methodologies." This article explains that literature reviews can cover a wide range of topics with varying levels of completeness and thoroughness based on literature analysis. These reviews may include research findings; the search method may or may not involve an exhaustive search; they may or may not include quality assessment; their synthesis is typically narrative; and the analysis can be chronological, conceptual, thematic, etc.

This is also reflected in the methodology with the phrase (highlighted in red in the text):

“A literature review was chosen as it allows for a broader and more diverse identification of available literature, as well as a more flexible and contextual exploration [22]. To achieve the aims of this review [22], a literature search was conducted to answer the question: How was the Comprehensive Gerontological Assessment (CGA) and its assessment tools implemented in Latin America and the Caribbean before, during, and after the COVID-19 pandemic?”

[22] Grant M, Booth A. A typology of reviews: an analysis of 14 review types and associated methodologies. Health Info Libr J 2009;26:91–108. https://doi.org/10.1111/J.1471-1842.2009.00848.X.

Additionally, and following the reviewer's recommendations, the inclusion and exclusion criteria, as well as the steps taken to conduct this literature review, have been added to the methodology section of the article with the following phrase (highlighted in blue in the text):

“Scientific articles were included if they contributed to answering the objectives of the literature review in conceptual and contextual terms or if they determined the validity, reliability, and/or provided reference values for any of the assessment tools in the countries of Latin America and the Caribbean. This could involve studies focusing exclusively on populations aged 60 years and older or as part of the general population.

Additionally, government documents and technical guidelines from countries in the region were included, whether they focused specifically on older populations or the general population. These documents needed to include any of the tools identified as valid and/or reliable in this review or propose CGA batteries for older adults at different levels of care, programs, and/or social-health services for older adults in countries in the Latin America and Caribbean region. 

Abstracts from conferences, scientific congress reports, dissertations, research re-ports, theses, letters to the editor, editorials, and commentaries were excluded.

The following steps were followed in the literature review process [23], depending on the type of document. In the case of scientific articles, the steps were: strategy-based search, article identification, duplicate elimination, title and abstract screening, full-text reading, selection, compilation into tables, analysis, interpretation of results and discussion, and referencing. Using the list of assessment tools for which validity, reliability, and/or reference values were determined, the review process proceeded to government documents from countries in the region. In the case of government documents, the steps were: search, identification, full-text reading focused on identifying the selected tools and CGA battery proposals mentioned in the document, selection, compilation into tables, analysis, interpretation of results and discussion, and referencing. A reference librarian, part of the research team, collaborated throughout the various stages of the literature review process.

A total of 800 scientific articles and 40 government documents were initially identified for the literature review. There were 254 duplicates, which were eliminated. After applying inclusion and exclusion criteria, 129 documents were finally selected for the literature review (110 scientific articles and 19 government documents). Figure 1 provides a diagram of the article selection process for the literature review.”

Along with this, bibliographic reference number 23 has been added to support the steps followed in the literature review:

[23] Chigbu U, Atiku S, Du Plessis C. The Science of Literature Reviews: Searching, Identifying, Selecting, and Synthesising. Publications 2023;11:2. https://doi.org/10.3390/PUBLICATIONS11010002.

We have also added a figure (Figure 1: Diagram of the article selection process for the literature review, categorized by type of document and country, highlighted in blue in the text) that illustrates the flow of eligible, excluded, and selected articles and documents for the literature review. These are further categorized by quantity, type of document, and country of origin, directly relating to the bibliographic references of the submitted scientific article.

Specifically, in Table 2, and following the bibliographic referencing standards, the numbering corresponds to the scientific article or government document that has proposed a CGA battery at some level of care, program, or social-health service for older adults in the context of Latin America and the Caribbean, which we have considered and adapted in light of the presented literature review and the authors' expertise.

Additionally, in Tables A1, A2, A3, and A4, and following the bibliographic referencing standards, the numbering corresponds to the scientific article that reflects the validation, determination of reliability, and/or use in a government document of the evaluation tool in a Latin American or Caribbean country.

Consequently, the literature review aligns with the objectives of our research and supports our broader discussion and analysis. Our literature review methodology was designed to ensure rigor and transparency. Searches were conducted in relevant databases using predefined search terms, with inclusion and exclusion criteria and explicit steps for the review (now more clearly outlined), and the findings were synthesized. Although we did not adopt the formal framework of a scoping review, the process we followed is robust and sufficient to address the research questions and objectives of this paper.

Comment:

2) With the inclusion of COVID-19 you bring a whole new dimension to the paper. I would strongly recommend that you separate the issues. Namely, you can do a separate paper for another journal on the gerontological assessment practices that were necessary during COVID-19. Keep the focus of the present paper purely on the assessment practices during 'normal' circumstances.

Response: Thank you. One of our research objectives was to analyze the adaptations made to CGA during the COVID-19 pandemic in the region. This stems from the fact that during the COVID-19 pandemic, professionals, care center teams, and governments were compelled to quickly adapt the implementation of CGA and its evaluation tools, often without significant prior experience, to ensure continuity of care for older adults, their families, and caregivers. Telehealth became the alternative.

This was an opportunity to advance CGA through Telehealth as an additional modality of care, projecting its use in normal situations, as had already begun to develop on a small scale before the pandemic. Therefore, to further support the need to explore this aspect, we have added the following phrase to the introduction (highlighted in blue):

“The CGA adaptations that were implemented during the pandemic and involved the use of telehealth could provides valuable lessons for employing this modality of care in regular and non-emergency situations as well."

In this context, the literature on Telehealth and CGA in the region is still in its early stages, and we believe that the findings gathered in this literature review provide a context, a modern emphasis on the article, and suggest new perspectives for innovation in normal contexts.

Therefore, considering the normal situations that the teams caring for older adults in the countries of Latin America and the Caribbean face—such as natural disasters, which are very frequent in the region (earthquakes, volcanic eruptions, tsunamis, hurricanes, among others); rurality (due to the region's geographical and ethnic diversity, from north to south, coast to mountains, plains to valleys, continent to islands); and regular situations older adults face, such as disabilities and care needs—these all limit social mobility and/or restrict the social-health participation of older adults. This is why, in section 4.4.4. CGA considerations from an innovation perspective, we have included the phrase that refers to the transfer of CGA experiences in Telehealth modality during the COVID-19 pandemic to regular normal situations (highlighted in red in the text):

“These strategies are not only useful in a pandemic context but also in situations of natural disasters, rural settings, disability, and caregiving, which limit social mobility and/or restrict the socio-health participation of OA.”.

This also answers our research objective.

Comment:

In terms of the layout the following:

1) It was noted that you often critique assessment instruments and practices while you are busy outlining the results of the study. Please keep the results section focused on what the literature indicates.

Response: Thank you. To address this point and the next two points, and incorporating your recommendations, we have added the title "4. Discussion" before the title "Challenges and opportunities for Comprehensive Gerontological Assessment in Latin America and the Caribbean," considering that, after a thorough reflection by the authors, this section is more focused on discussing considerations, projections, and challenges, while also offering some constructive critiques on the topic analyzed in this literature review.

For this reason, the titles in this section have also been renumbered, now appearing as follows (highlighted in blue in the text):

  1. Discussion

4.1. Challenges and opportunities for Comprehensive Gerontological Assessment in Latin America and the Caribbean

4.1.1 CGA considerations from an administration perspective tools

4.1.2. CGA considerations from an academic training perspective

4.1.3. CGA considerations from a research perspective

4.1.4. CGA considerations from an innovation perspective

4.1.5. CGA considerations from a management perspective

Additionally, this has led to the renumbering of the Conclusion, now appearing as (highlighted in blue in the text): "5. Conclusions."

Comment:

2) Include a section after the results called 'Discussion' - here you can debate the assessment tools and include critique where warranted.

Response: Thank you. We have included a section on “Discussion”, as mentioned in the previous point.

Comment:

3) In the conclusion section you can add 'recommendations' for future research, practice, training, continiuning professional development, policy, etc.

It is not helpful to 'mix' all these 'key ingredients' of a paper under the one heading related to 'Results'.

Response: Thank you. We have addressed this point in our previous reply.

Additionally, the following phrase has been added to the conclusion (highlighted in blue in the text):

“Consequently, the Comprehensive Gerontological Assessment (CGA) batteries pre-sented in this literature review establish a minimum standard for implementation in care models and public policies for the countries of Latin America and the Caribbean”

Comment:

Also see attached the manuscript with some additional comments.

Response: Thank you. We have carefully reviewed the attached file, and the indicated adjustments have been made to improve the layout of the article. Similarly, the specified paragraphs have been reviewed, and the requested changes have been made.

Another comment on the platform:

The entire manuscript will have to be language edited. In some instances one sentences is the length of an entire paragraph. A reader looses comprehension when sentences are extra long. The writing must be clear, focused and word economic.

Response: Thank you. We have reviewed the entire article in terms of writing, and in the longer paragraphs, a comma or a period has been added as appropriate to improve the readability of the text and provide clear and precise ideas to the reader.

Comment:

I trust these comments will be helpful to improve your paper.

Response: Thank you very much for your valuable comments, which allow us to deliver an even more strengthened scientific article for publication in the journal.

Reviewer 3 Report

Comments and Suggestions for Authors

Thank you for providing the reviewer an opportunity to evaluate this manuscript, it was interesting. The reviewer has provided the following suggestions and comments.

·      The reviewer admires the authors for their thorough review and categorization of the data; however, the authors assert in the abstract that there is insufficient training for geriatricians and health professionals specializing in geriatrics, attributing this deficiency to a gap in older adults assessment training. The reviewers are unclear regarding the location in the introduction where the authors have presented the evidence.

·      The authors utilize the term CGA to denote both the comprehensive geriatric assessment and the comprehensive gerontology assessment. The reviewer suggests that, due to the common usage of the abbreviation CGA to denote comprehensive geriatric assessment, the authors should consider employing an alternative acronym for comprehensive gerontological assessment.

·      What is the quantity and percentage of research findings on assessment methods comprehensive gerontological assessment, comprehensive geronto-geriatric assessment, and comprehensive geriatric assessment as indicated in the author's study?

·      The reviewer believes that most geriatricians and healthcare workers are knowledgeable about and utilize comprehensive geriatric assessment. The reviewer suggests that the literature would be enhanced if the authors detailed the similarities or differences between the comprehensive gerontology assessment and comprehensive geriatric assessments in the results section.

·      Are there limitations to this study?

Author Response

Response to Reviewer 3

Comment:

Thank you for providing the reviewer an opportunity to evaluate this manuscript, it was interesting. The reviewer has provided the following suggestions and comments.

Response: Thank you.

Comment:

The reviewer admires the authors for their thorough review and categorization of the data; however, the authors assert in the abstract that there is insufficient training for geriatricians and health professionals specializing in geriatrics, attributing this deficiency to a gap in older adults assessment training. The reviewers are unclear regarding the location in the introduction where the authors have presented the evidence.

Response: Thank you for your comment. To reply to your comment, we have highlighted in red colour the corresponding phrases of the introduction (page 2, lines 58 - 77).

Comment:

The authors utilize the term CGA to denote both the comprehensive geriatric assessment and the comprehensive gerontology assessment. The reviewer suggests that, due to the common usage of the abbreviation CGA to denote comprehensive geriatric assessment, the authors should consider employing an alternative acronym for comprehensive gerontological assessment.

Response: Thank you for your comment. In Spanish, the acronym CGA is used interchangeably to refer to both terms (VGI: Valoración Geriátrica Integral AND Valoración Gerontológica Integral). Also, in studies that do employ the term Comprehensive Gerontological Assessment, the same acronym – CGA – is used. Basically, in both languages, the same acronym is used for both terms. As the term “Comprehensive Geriatric Assessment” is used only a few times (only once) in the manuscript, we think it would be better to use CGA for “Comprehensive Gerontological Assessment”.

Comment:

What is the quantity and percentage of research findings on assessment methods comprehensive gerontological assessment, comprehensive geronto-geriatric assessment, and comprehensive geriatric assessment as indicated in the author's study?

Response: Thank you for your comment. The concept of Comprehensive Geriatric Assessment (CGA), in the context of geriatrics, has evolved and transitioned in Latin America and the Caribbean to the concept of Comprehensive Gerontological Assessment (CGA), within the context of gerontology, with an intermediate stage involving the Comprehensive Geronto-Geriatric Assessment (CGGA).

This transition has been reflected in the use of the Comprehensive Geronto-Geriatric Assessment concept, primarily in public policies of some countries in the region, as identified in this literature review, and explicitly stated in the following phrase from the results (highlighted in red in the text):

“This concept has got closer to gerontology (with a biopsychosocial, broad, and holistic vision) and has transitioned to the concept of Comprehensive Geronto-Geriatric As-sessment [24–26], which has already been used in some public policy programs [24–26].”

At the same time, the concept of Comprehensive Gerontological Assessment has been applied in various postgraduate programs, conferences, public policies, and research (in both Spanish and English) in Latin American and Caribbean countries, as identified in this literature review, and explicitly stated in the following phrase from the results (highlighted in red in the text):

“This concept is not new and has been used before, for example, at the XXV Inter-national Congress of the Galician Society of Gerontology and Geriatrics in 2013 in Spain [28]. Furthermore, it has been incorporated into postgraduate programs aimed at physi-cians, health professionals, and psychosocial workers, such as the Master’s in Geron-tological and Geriatric Kinesiology at Universidad San Sebastián in Chile, and the Doctorate in Gerontological Research at Universidad Maimónides in Argentina. Addi-tionally, it has been used in recent scientific articles in both Spanish [3,29,30] and Eng-lish [31,32], and in resolutions by the Ministry of Health of Argentina [33].”

Additionally, to provide more detail in the methodology, we have included a figure showing the flow of eligible, excluded, and selected articles and documents for the literature review, categorized by quantity, type of document, and country of origin. This figure directly relates to the references of the submitted scientific article and is secondary to the search strategy used. This new figure is titled, Figure 1: Diagram of the article selection process for the literature review, categorized by type of document and country (highlighted in blue in the text).

This new figure is associated with the following phrase in the methodology (highlighted in blue in the text):

“A total of 800 scientific articles and 40 government documents were initially iden-tified for the literature review. There were 254 duplicates, which were eliminated. Af-ter applying inclusion and exclusion criteria, 129 documents were finally selected for the literature review (110 scientific articles and 19 government documents). Figure 1 provides a diagram of the article selection process for the literature review.”

Comment:

The reviewer believes that most geriatricians and healthcare workers are knowledgeable about and utilize comprehensive geriatric assessment. The reviewer suggests that the literature would be enhanced if the authors detailed the similarities or differences between the comprehensive gerontology assessment and comprehensive geriatric assessments in the results section.

Response: Thank you for your comment. Considering the response to your first comment (above), and as stated in the introduction, there is a gap in undergraduate and postgraduate education and training for physicians and professionals in Latin America and the Caribbean on topics related to Geriatric Syndromes and Comprehensive Gerontological Assessment. Additionally, there is a shortage of geriatricians in the countries of the region. Therefore, we cannot assert that the majority of geriatricians and those working with older adults are familiar with comprehensive assessment and its evaluation tools in the countries of the region.

Additionally, at the beginning of the first paragraph of the results, in section 3.1, it has been emphasized that the Comprehensive Geriatric Assessment has a biomedical, reductionist, and geriatrized view of the social and healthcare services for older adults, adding this idea at the end of the sentence (highlighted in blue in the text):

“Traditionally, from the field of Geriatrics and through the expertise of geriatri-cians, the concept of Comprehensive Geriatric Assessment was initially proposed and used. This type of assessment – with its classic four domains: biomedical or clinical, functional, mental, and socio-familial [5,6] – has a biomedical, reductionist, and geri-atrized view of the social and healthcare services for older adults.”

At the same time, it has been added in the second paragraph of the results that gerontology has a biopsychosocial, broad, and holistic view (highlighted in blue in the text):

“This concept has got closer to gerontology (with a biopsychosocial, broad, and holistic vision) and has transitioned to the concept of Comprehensive Geronto-Geriatric As-sessment [24–26], which has already been used in some public policy programs [24–26].”

This establishes points of divergence and complementarity between the science (Gerontology) and its branch (Geriatrics), as defined in the introduction (highlighted in red in the text):

“In this context, the science dedicated to studying aging and the elderly population is gerontology [3], which addresses all aspects of OA, their families, and their envi-ronment, drawing from multiple and diverse disciplines and professions [3]. Geriat-rics, on the other hand, is the science branch focused on addressing the potentialities, needs, problems, and geriatric syndromes of OA at different levels of healthcare [4].”

And consequently, between the Comprehensive Geriatric Assessment and the Comprehensive Gerontological Assessment (the latter as an expanded and updated concept), in light of the particularities of Latin America and the Caribbean, as outlined in the following paragraph of the results (highlighted in red in the text):

“However, considering the geographic, demographic, and ethnic diversity (from north to south, from coast to mountain ranges, from plains to valleys, from continent to is-lands) that characterizes Latin America and the Caribbean, the significant progress in gerontology [3], the diversity of university professions that address OA in the region (some of which are unique and present only in certain countries, such as gerontolo-gists, psychomotricians, and music therapists), and the variety of contexts in which interventions with OA take place (ranging from hospital to community, urban and ru-ral), this concept should definitively transition to the concept of “Comprehensive Gerontological Assessment”, proposal we make in this literature review. CGA should also be understood as one of the three main pillars or cornerstones of gerontology and geriatrics, along with interdisciplinary work and level-based coordination and management [27].”

Comment:

Are there limitations to this study?

Response: Thank you. We have included two limitations of our research in the discussion section with the following phrase, highlighted in blue in the text:

“One of the limitations of this study is that it did not assess which CGA tool is most commonly used in each country; it identified instead which tools are used or recom-mended in each country based on government documents. Furthermore, no govern-ment documents were found concerning the utilization of CGA tools in countries with an incipient demographic transition and low rates of population aging.”

Thank you very much for your valuable comments, which allow us to deliver an even more strengthened scientific article for publication in the journal.